# 3D Ultrasound Reconstructions of the Carotid Artery and Thyroid Gland Using Artificial-Intelligence-Based Automatic Segmentation—Qualitative and Quantitative Evaluation of the Segmentation Results via Comparison with CT Angiography

**DOI:** 10.3390/s23052806

**Published:** 2023-03-03

**Authors:** Tudor Arsenescu, Radu Chifor, Tiberiu Marita, Andrei Santoma, Andrei Lebovici, Daniel Duma, Vitalie Vacaras, Alexandru Florin Badea

**Affiliations:** 1Computer Science Department, Technical University of Cluj-Napoca, 400114 Cluj-Napoca, Romania; 2Chifor Research SRL, 400068 Cluj-Napoca, Romania; 3Department of Preventive Dentistry, “Iuliu Hatieganu” University of Medicine and Pharmacy, 400083 Cluj-Napoca, Romania; 4Radiology, Surgical Specialties Department, “Iuliu Hatieganu” University of Medicine and Pharmacy, 400006 Cluj-Napoca, Romania; 5Radiology and Imaging Department, Cluj County Emergency Clinical Hospital, 400006 Cluj-Napoca, Romania; 6Department of Neurosciences, “Iuliu Hatieganu” University of Medicine and Pharmacy, 400012 Cluj-Napoca, Romania; 7Neurology Department, Cluj County Emergency Hospital, 400012 Cluj-Napoca, Romania; 8Anatomy and Embryology, Faculty of General Medicine, “Iuliu Hatieganu” University of Medicine and Pharmacy, 400006 Cluj-Napoca, Romania

**Keywords:** carotid artery ultrasonography, medical imaging, artificial intelligence, carotid disease diagnosis, atherosclerosis diagnosis

## Abstract

The aim of this study was to evaluate the feasibility of a noninvasive and low-operator-dependent imaging method for carotid-artery-stenosis diagnosis. A previously developed prototype for 3D ultrasound scans based on a standard ultrasound machine and a pose reading sensor was used for this study. Working in a 3D space and processing data using automatic segmentation lowers operator dependency. Additionally, ultrasound imaging is a noninvasive diagnosis method. Artificial intelligence (AI)-based automatic segmentation of the acquired data was performed for the reconstruction and visualization of the scanned area: the carotid artery wall, the carotid artery circulated lumen, soft plaque, and calcified plaque. A qualitative evaluation was conducted via comparing the US reconstruction results with the CT angiographies of healthy and carotid-artery-disease patients. The overall scores for the automated segmentation using the MultiResUNet model for all segmented classes in our study were 0.80 for the IoU and 0.94 for the Dice. The present study demonstrated the potential of the MultiResUNet-based model for 2D-ultrasound-image automated segmentation for atherosclerosis diagnosis purposes. Using 3D ultrasound reconstructions may help operators achieve better spatial orientation and evaluation of segmentation results.

## 1. Introduction

Atherosclerosis is a progressive disease characterized with accumulation of lipids and fibrous elements in the large arteries. It initiates in early adulthood and manifests clinically in middle to late adulthood after decades of plaque progression. Extracranial atherosclerotic disease (ECAD), primarily carotid artery stenosis, accounts for approximately 18–25% of ischemic stroke [1], and over one-third of all strokes are caused by thromboembolism from a stenotic carotid artery [2]. In patients with acute ischemic stroke, the prevalence of intracranial atherosclerosis does not differ between women and men, while extracranial atherosclerosis is less often present in women compared with in men [3]. European and U.S. guidelines for prevention of stroke in patients with carotid plaque are based on quantification of the percentage reduction in the luminal diameter due to the atherosclerotic process in order to select the best therapeutic approach [4]. Stenosis quantification is not sufficient, and plaque characterization is also necessary when discussing treatment strategies and prognosis. The accuracy of carotid artery stenosis (CAS) diagnosis has substantially increased over the decades, with the progressive technological developments passing from measurement of the narrowing carotid artery diameter to evaluation of the increased velocity field near the obstruction/lesion site in the carotid artery, with increased emphasis now on detection of additional parameters to characterize plaque vulnerability [5]. Early and secure identification of these plaques would allow development of individualized therapeutic and pharmacological strategies, applied in a timely manner [6]. Soft plaque, plaque ulceration, and increased common carotid artery wall thickness on computed tomography angiography (CTA) are associated with ipsilateral cerebrovascular ischemia, while calcified plaque is associated with downstream ischemic events to a lesser extent in comparison [7]. Additionally, for other large blood vessels, with computed tomography coronary angiography (CTCA), high-risk plaque (HRP) lesions with obstructive (>50%) stenosis and large, low-attenuation plaque areas, which demonstrated great association with future acute coronary syndrome (ACS), were identified, while stenosis severity alone in the absence of HRP features was not associated with future ACS [8].

Today, various diagnostic modalities are available for evaluation of carotid artery disease: color Doppler ultrasonography, computed tomography angiography, magnetic resonance angiography, and intra-arterial digital subtraction angiography (DSA) [9]. Other potential imaging tools are optical coherence tomography (OCT), photoacoustic tomography (PAT), and infrared (IR) thermography [5]. DSA is still considered a gold standard in assessment of stenosis, but because of the stroke risk, patient discomfort, and high cost, it is increasingly being replaced with noninvasive techniques [10]. Assessment of plaque enhancement is limited in the case of single-phase CTA, and multiphase CTA is rarely performed outside of research studies due to radiation concerns [11].

The aim of our study is to evaluate the feasibility of a noninvasive and low-operator- dependent imaging method for carotid-artery-stenosis diagnosis. A previously developed prototype for 3D ultrasound scans based on a standard ultrasound machine and a pose reading sensor was used for this study. Working in a 3D space and processing data using automatic segmentation lowers operator dependency. Additionally, ultrasound imaging is a noninvasive diagnosis method. Artificial intelligence (AI)-based automatic segmentation of the acquired data was performed for the reconstruction and visualization of the scanned area: the carotid artery wall, the carotid artery circulated lumen, soft deposits, and calcified deposits. The evaluation was carried out via comparing the result with CT angiographies of healthy and carotid-artery-disease patients. The chosen AI model was a variation of the UNet CNN (convolutional neural network) model, which has yielded good results in other studies [12,13,14].

## 2. Materials and Methods

This study was conducted in accordance with the Declaration of Helsinki and approved by the Institutional Review Board (or Ethics Committee) of the Iuliu Hatieganu University of Medicine and Pharmacy, Cluj-Napoca, Romania (1NZ202/11 July 2022). Informed consent was obtained from all subjects involved in this study.

### 2.1. Ultrasound Data Acquisition

A 3D high-frequency ultrasound (US) imaging prototype based on a standard medical 2D US scanner (Vinno 6, Suzhou, China) with a high frequency (10–23 MHz), a small-aperture (12.8 mm) linear transducer (X10-23L, Vinno, Suzhou, China), and an articulated measurement arm (Evo 7, RPS Metrology (Sona(VR)/Italy)), used as a pose-reading sensor, was used for this study. The linear array transducer was used at 20 MHz, having as its setup a standard vascular preset and a reference depth of 2.5 cm. The probe was attached to the sensor, and the scanning was performed after temporal and spatial calibration of the devices. This prototype had been developed previously for periodontal-tissue ultrasound imaging investigations, as described in other studies [15,16] detailing the technical performances and obtained results for periodontal-tissue 3D ultrasound reconstructions. Using the same prototype and setup, with the added vascular preset of the ultrasound machine, the carotid arteries in the neck region and the thyroid were scanned. Four consecutive scans of 2 conscious and responsive patients, hospitalized for stroke signs and symptoms, and 1 healthy patient were performed by the same radiologist. The examination began with the patient in a supine position and with the head in a slight hyperextension. The transducer was placed in the right submandibular area in the region of the carotid bulb in a transverse position; the examination continued in craniocaudal motion in the lateral cervical (jugular–carotid) region, examining the right common carotid artery in its cervical pathway until the right lobe of the thyroid gland was reached, then continuing examining the entire thyroid gland, starting with the right lobe, moving to the isthmus and finally examining the left lobe. After that, the left common carotid artery was examined from the level of the left thyroid lobe in caudocranial manner until the left carotid bulb was reached. The examination followed a “U”-letter pattern from the right to the left side.

The examination was performed with the craniocaudal and midsagittal free movement of the operator’s hand, which held the transductor and the pose-sensor assembly, with irregular operator-dependent movement speed, along the previously described trajectory. Because of the free hand movement of the transductor and the pose-sensor assembly, the number, relative spacing, and orientation of the 2D frames in 3D space, belonging to distinct scans of the same examined region of the same patient, differed along the scan direction.

### 2.2. CT Angiography

The two patients who had stroke signs and symptoms benefited from the standard examination for their condition, performed on a General Electrical (GE) Revolution EVO 128-slice CT system with a standard angiogram supra-aortic trunk protocol. The source parameters were set at 120 KV and 50 mA for the CT angiography. Each patient was positioned, supine, in the gantry, with their arms at their side. First, the scout image was acquired from the midchest to the head vertex, with a scanning extent from the aortic arch to the vertex and a scan direction orientated caudocranially as per a noncontrast study performed prior to the angiographic phase. The contrast medium was injected in the peripheric cubital vein at around a dose of 80–90 mL of Omnipaque contrast agent, with 100 mL of saline chaser at around 4.5 mL/s, with a set region of interest (ROI) in the descending aorta and automatic bolus tracking with minimal delay and scan triggering when 120 HU was reached in the descending aorta. Diagnostically, every major vessel from the aortic trunk upward, including the intracranial arteries, was assessed, with the types of plaque (calcified, soft/mixed) and grading stenosis reported according to five categories: normal (0% stenosis), mild stenosis (1–49%), moderate stenosis (50–74%), severe stenosis (75–99%), and occluded artery.

### 2.3. Original 2D Ultrasound Image Segmentation

After acquisition, the 2D neck-region US images were extracted using an in-house-developed Python software application and were saved as png files. Semiautomatic segmentation was performed by a student trained and supervised by an experienced radiologist. Five anatomical elements (the carotid artery wall, the circulated lumen, soft plaque, calcified plaque, and the thyroid) were searched for and identified if present (Figure 1). Every anatomical element was masked using an in-house-developed semiautomatic software annotation tool relying on a customized region-growing-based segmentation algorithm, described in a previous study [16]. The algorithm expects the user to click on seed points, which are “grown” via iteratively adding neighboring pixels with similar intensities. The algorithm is customized in such a way that already-labeled pixels are not considered. The similarity predicate is controlled with a threshold, T, that is tunable by the user. It interacts with the user through the keyboard and the mouse as well as through the OpenCV and PyQT GUIs to display input and output images and to control a track-bar used to set the threshold, T, for region growing. Tools such as the eraser and the pencil-drawing tool were used for fine corrections.

### 2.4. CT-Scan Segmentation Methods

For manual segmentation of CBCT, the software Slicer 3D, version 5.2.1 [17], was used.

#### 2.4.1. Semiautomatic Segmentation Method of Patient CT Scans Using 3D Slicer

The following steps were taken to semiautomatically segment the reference CT scans for the patients in the scope of this study:Patient’s DICOM scans were loaded into 3D Slicer.The CT angiography volume was selected as the closest 3D representation of the regions of interest: the carotid circulated lumen, the carotid artery wall, carotid artery soft plaque and calcified plaque, and the thyroid gland.The rendering mode was adjusted to MR angio to obtain a colored view of the bones and the blood vessels in the volume and identify the regions of interest.The regions of interest were selected to capture both the carotid arteries and the entire neck length of the patient, and the volume was cropped to the space inside the regions of interest.Segments for the relevant anatomical regions of the volume were created using the segmentation editor: for the carotid circulated lumen, the carotid artery wall, carotid artery soft plaque and calcified plaque, and the thyroid gland.Segmentation editor tools such as region growing, the paint tool, the eraser tool, the islands tool, etc. were used to segment various tissue types across various DICOM frames on all 3 axes. At the end, the 3D rendering feature of the region-growing tool was used to visualize and fine-tune the result before it was exported to STL format in order to have it as a comparison reference for the ultrasound segmentation. The segmentation was conducted in such a way so that the segmented result would overlap as closely as possible to the MR-angio visualization of the angiography volume, described above, this visualization being considered the gold-standard 3D representation of tissues of interest.

#### 2.4.2. Methodology for CT Segmentation of Ground Truth

The circulated lumen was segmented from the CT angiography volume (Figure 2).
It was possible for some of the hard deposits to be partially or totally included in the circulated-lumen segmentation because the HU range for these, in some cases, depending on deposit density, was similar to the HU range of the contrast substance.The intersection volumes between the segmented circulated lumen on the CT angiography volume and the segmented hard deposits on the native CT volume were excluded from the result. This represented the ground truth for the circulated lumen.Hard deposits were segmented from the native CT volume. This was the ground truth.The thyroid was segmented from either the native or the CT angiography volume.

### 2.5. Machine-Learning Dataset Preparation

For the construction of the machine-learning training dataset, five separate 3D ultrasound scans, acquired using the abovementioned prototype, were selected from the totality of those performed by an experienced radiologist, resulting in the image datasets described in Table 1. The acquired original ultrasound 2D images in each selected dataset were segmented by a student trained and supervised by an experienced radiologist. The outcome of these actions was the sets of masks corresponding to the original 2D US images, from the datasets in Table 1 that were in the scope of the segmentation. The distribution of the number of masks per class in the datasets is presented in Figure 3. The dataset was divided into sections for training, validation, and testing according to the 0.8–0.1–0.1 ratio.

### 2.6. Ultrasound-Scan Automatic Segmentation Methods

The automatic segmentation of the images was performed with pixel-level classification of the original US images in the abovementioned 5 classes of interest (the blood vessel wall, the circulated lumen of the blood vessel, soft plaque, calcified plaque, the thyroid gland) and the background. For that purpose, MultiResUNet [18] CNN architecture was used. This is an improvement of the original UNet architecture that solves the scale problem of tissues of interest found in medical images via replacing the simple convolutional blocks with MutiRes blocks consisting of the concatenation of three successive 3 × 3 convolutions and additional input. Another improvement is the replacement of the skip connections with residual paths consisting of 4 successive blocks, each performing a 3 × 3 convolution with additional input. The predictions were made for 5025 2D original US images selected from scans 2, 3, and 4 from the two carotid-disease patients (Table 2).

### 2.7. 3D Ultrasound Reconstructions of the Carotid Arteries

Methodology for the ultrasound reconstruction:

Using the 3D US scanner prototype and the developed software, after the US data was acquired, each frame was paired or matched with the sensor’s readings. The spatial coordinates and orientation of each frame were determined.

The acquired 2D frames were 375 × 735-pixel grayscale png files. For a subset of the acquired 2D frames, a manual segmentation of ground-truth masks for five tissue categories (the carotid wall, the carotid circulated lumen, carotid calcified plaque, carotid soft plaque, the thyroid) was performed. These ground-truth masks were combined into multiple training datasets for a MultiResUNet [18] convolutional neural network model. Later, the trained model was used to automatically segment the 5 tissue classes previously mentioned, plus the background, on a selected set of acquired ultrasound 2D frames. The resulting segmented masks were used to make 3D reconstructions of the morphologies for the abovementioned tissue classes for the selected set of ultrasound acquisitions. This was achieved through having all 2D ultrasound frames reorganized in the 3D space and masked with the results from the semiautomatic or automatic segmentation, thus reconstructing the 3D data volume.

For the training and segmentation, we used a dedicated machine with the following specifications: GPU GeForce RTX 2060 8GB RAM, Nvidia driver 450.51.06, CUDA version 11.0, and TensorFlow 2.0.

### 2.8. Qualitative Analysis of the 3D US Reconstructions

Methodology for the qualitative evaluation of the 3D volumes obtained from the CT and the ultrasound for ground-truth segmentation classes from the CT:

For the qualitative comparison, the ultrasound 3D reconstructions, based on automatically segmented 2D ultrasound frames, were visually compared with reference CT scans of the same patients. The focus of the comparison was on the circulated lumen and calcified plaque.

A refinement applied to the qualitative visual inspection of the USs and the corresponding CT scans was the visual comparison of the circulated-lumen center line identified in the CT and US scans selected for this study.

Segmentation of the CT DICOMs in scope:

For this study, CT scans of 2 patients with clinically confirmed atherosclerosis were used. For each patient, segmentation of the circulated lumen and calcified plaque deposits was performed using 3D Slicer [19] on a CT scan selected for this study. Additionally, for CT scans in this study, a calculation of the center line of the circulated lumen was performed.

The main 3D Slicer toolbox item used for segmentation was the Segmentation Editor augmented with the Segment Editor Extra Effects and Slicer VMTK plug-ins. The Segment Editor Extra Effects plug-in was used for segmenting the circulated lumen. The Slicer VMTK plug-in was used for calculating the center line of the circulated lumen.

### 2.9. Quantitative Analysis of the 3D US Reconstructions

For quantitative analysis of the 3D US reconstructions, point-cloud mathematical comparison methods were employed. The tool used for this purpose was Cloud Compare Open-Source Software 2.12.4 (CCOSS).

The method used was to select, for each patient in the study group, a reference 3D ultrasound reconstruction based on manually segmented masks for the 5 tissue classes described above. This reference was then compared with each 3D US reconstruction based on masks that had been automatically segmented with the specific machine-learning models described above. The first step consisted of alignment of the corresponding point clouds of the two reconstructions, with the help of Cloud Compare’s point-cloud-alignment tool. The next step was fine-tuning the alignment using Cloud Compare’s fine registration tool. The last step was to measure the distance between the aligned point clouds.

The metrics used in the comparison were the mean distance between the point clouds, the standard deviation of the alignment, the scale factor of the alignment, theoretical overlap, and the root mean square (RMS).

## 3. Results

### 3.1. 2D Automatic Segmentation

#### 3.1.1. Automatic Segmentation Results Compared with the Gold Standard (Operator’s Segmentation)

The qualitative analysis of the results of the automatic segmentation/prediction of a US image using the CNN model (MultiResUNet) can be seen in the example shown in Figure 4.

#### 3.1.2. Automatic Segmentation (MultiRes U-Net) Training Results Metrics

The average intersection over union (IoU) [20] was 80.29% for the dataset formed of 3864 2D US images from the neck region, showing the carotid arteries and the thyroid gland. The best performances were achieved for the interior of the blood vessel, the circulated lumen segment: an IoU of 93.09% and a Dice coefficient (Dice) [20] of 0.97. Additionally, the accuracy, specificity, recall, and precision were calculated and are presented in Table 3.

### 3.2. 3D US Reconstruction Compared with CT Angiography

Three-dimensional ultrasound reconstructions were performed for easier spatial orientation for the operator during masks’ semiautomatic segmentation. The continuity of the artery lumen and wall was easily verified using these reconstructions. Qualitative analyses were performed for the 3D ultrasound reconstructions in comparison with the CT angiography scans as the gold standard (Figure 5).

### 3.3. Quantitative Analysis for the 3D US Reconstruction Based on Automated Mask Segmentation

The 3D ultrasound reconstructions for scans 2, 3, and 4 for the two patients with atherosclerosis were performed based on the automated segmentation of the ultrasound images. For quantitative evaluation of the results of the segmentation, the 3D US reconstructions were compared with scan 1 for the same patient. For scan 1, 3D ultrasound reconstruction manual segmented frames were considered masks. In Table 4, the mean distances and the standard deviations (stds) for the aligned 3D objects are as calculated using CCOSS. In Figure 6, one can visualize the distance between two aligned 3D ultrasound reconstructions. Red-colored points are further apart and blue ones are closer.

## 4. Discussion

The circulated lumen of the carotid artery had the most regulated form compared with the masks from the other classes. More than that, along with the arterial wall, it was the most present element in the 2D ultrasonographic images, followed by the thyroid. That explains the highest IoU and Dice values: 0.93 and 0.97, respectively. The average scores for all classes in our study were 0.80 for the IoU and 0.94 for the Dice. Groves et al., in their study segmenting the carotid artery from US images in 2020, obtained an IoU value that was a little higher and a lower Dice value: 0.88 for the IoU and 0.90 for the Dice [21]. They obtained the highest values using a Mask R CNN AI model for the automatic segmentation. Zhuang et al. [22] reported, for carotid intima, a 95% Dice value, using their own AI model based on a superpixel generation algorithm and the fractal theory for automatic segmentation; this is comparable with our result of 96%. Li et al., in 2022 [23], obtained, for a FRDD-Net model, the best results of automated carotid plaque ultrasound image segmentation compared with other CNN models: an average IoU of 78.18. In our study, for the MultiResUNet model, the IoU for the soft plaque segmentation was 75.5%. A study by Smits et al. [24] showed that agreement between manual and automated segmentation of plaque components, lipid-rich necrotic cores (LRNCs), and calcifications was poor despite good interscan reproducibility of both methods. In our study, with an IoU of 0.64, the automated segmentation of the calcified plaque obtained the lowest score. The sizes of the masks from this class and the small number of calcification masks from the dataset were probably the main cause of this poor result.

The qualitative evaluation of the 3D US reconstructions was performed with visual analysis comparing the shapes and number of deposits with the CT angiography and comparing the reconstructions based on AI-segmented masks with those manually segmented by the operator. The reconstructions showed good quality, excepting scan 2 from patient 2, where the shape of the vessel was irregular, probably due to a corruption of the acquired data from the pose-reading sensor.

Despite the limitations of our study due to the small number of patients and the reduced size of the dataset, especially for plaque classes, promising results were obtained. Future studies are necessary to fully demonstrate the feasibility of automated segmentation of anatomical elements for carotid-disease diagnosis. Larger datasets and their quantitative evaluations of the results via comparison with CT angiography are needed after an improved method for alignment and scaling of obtained 3D models is found.

## 5. Conclusions

The present study demonstrated the potential of the MultiResUNet CNN model for 2D-ultrasound-image automated segmentation for atherosclerosis diagnosis purposes. Using 3D ultrasound reconstructions may help operators with better spatial orientation and evaluation of the segmentation results. Future studies are needed, using larger datasets and quantitative evaluation methods of the results.

## Figures and Tables

**Figure 1 sensors-23-02806-f001:**
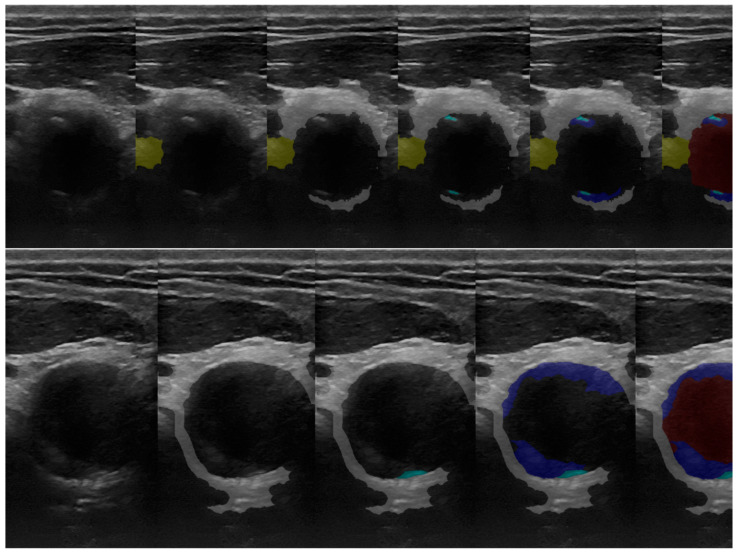
Manual and semiautomatic segmentation using our own software application: original US image, thyroid—yellow, carotid arterial wall—white, calcified plaque—light blue, soft plaque—dark blue, and circulated lumen—red.

**Figure 2 sensors-23-02806-f002:**
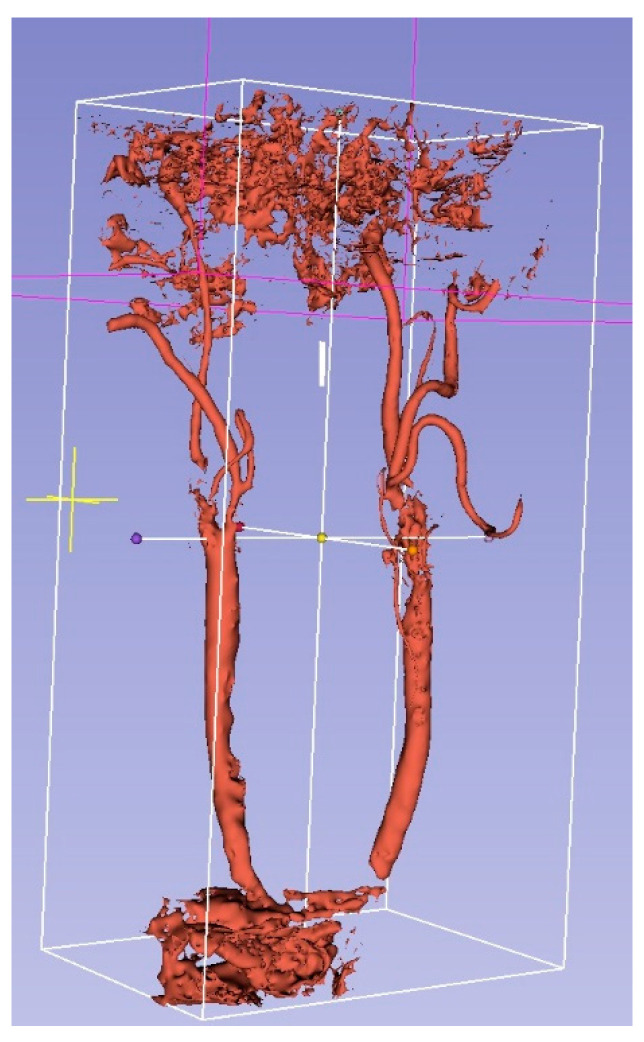
Segmented blood vessels from CT angiography.

**Figure 3 sensors-23-02806-f003:**
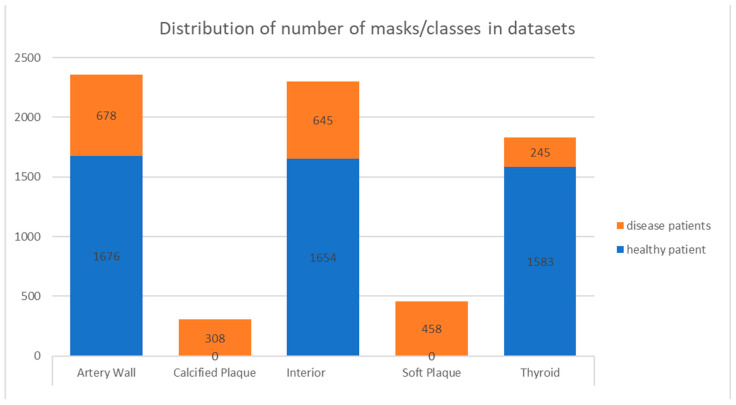
Distribution of number of masks/classes in datasets. Datasets from carotid-artery-disease patients—orange; healthy patients—blue.

**Figure 4 sensors-23-02806-f004:**
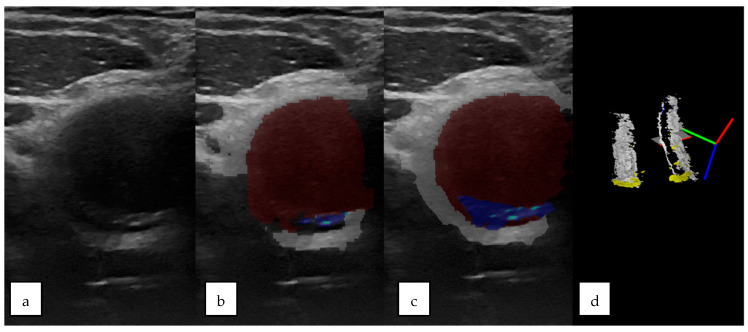
(**a**) Original US image and (**b**) CNN-based automatic segmentation. Arterial wall—white, calcified plaque—light blue, soft plaque—dark blue, circulated lumen—red. (**c**) Masks generated by the operator and (**d**) 3D US reconstructions of the scanned area, based on masks generated with CNN-based automatic segmentation.

**Figure 5 sensors-23-02806-f005:**
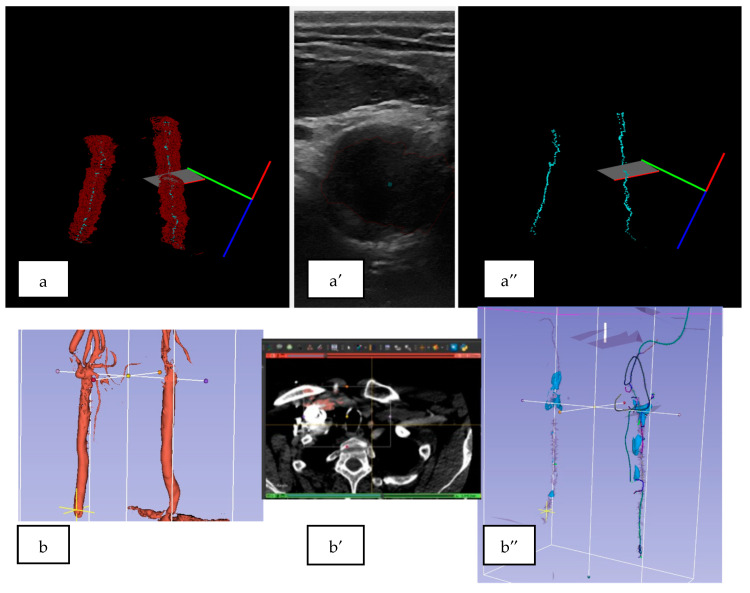
(**a**) The 3D US reconstruction of the carotid artery circulated lumen, (**a′**) the centroid of the cross-section of the circulated lumen, and (**a″**) representation in the 3D space of the centroids of the 2D ultrasonographic section through the carotid artery. (**b**) The CT angiography for the carotid artery of the circulated lumen, (**b′**) the transversal section through the carotid arteries’ CT scans, and (**b″**) extracted centroids of the carotid arteries, from the CT angiography represented in the 3D space.

**Figure 6 sensors-23-02806-f006:**
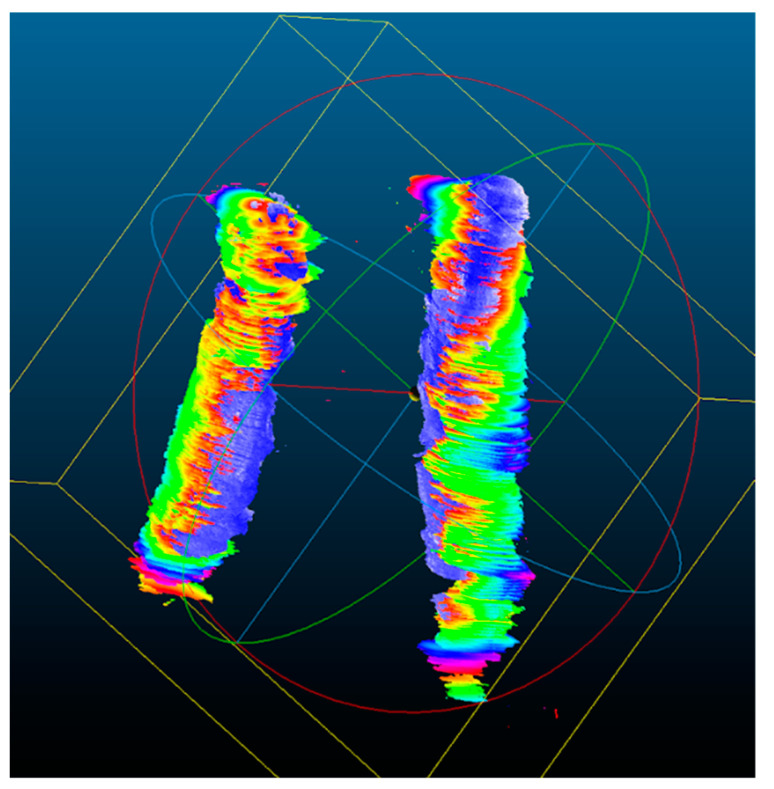
Patient 1, scan 2, circulated lumen evaluated with Patient 1, scan 1, circulated lumen as reference, aligned using CCOSS. The 100% overlapped 3D points are blue; the most distant points are represented in red.

**Table 1 sensors-23-02806-t001:** Dataset total frames per class.

Dataset	Number of Frames	Patient
Dataset 1	1001	Healthy (patient 1)
Dataset 2	971	Healthy (patient 1)
Dataset 3	959	Healthy (patient 1)
Dataset 4	531	Carotid disease (patient 2)
Dataset 5	222	Carotid disease (patient 3)

**Table 2 sensors-23-02806-t002:** The frames selected from the acquired US data for the carotid-artery-disease patients.

US Scan	Frame Interval	Number of Frames	Type of Segmentation
Healthy Patient Scan 1–4	-	2931	Manual
Carotid Disease Patient1 Scan 1	530–1060	531	Manual
Carotid Disease Patient1 Scan 2	330–1100	770	AI Prediction
Carotid Disease Patient1 Scan 3	230–1200	970	AI Prediction
Carotid Disease Patient1 Scan 4	280–1210	930	AI Prediction
Carotid Disease Patient2 Scan 1	400–622	222	Manual
Carotid Disease Patient2 Scan 2	330–1080	750	AI Prediction
Carotid Disease Patient2 Scan 3	280–1000	720	AI Prediction
Carotid Disease Patient2 Scan 4	310–1195	885	AI Prediction

**Table 3 sensors-23-02806-t003:** Prediction performance based on reference training data, where, for the given dataset and tissue class name, **tp** represents the number of true positive predictions, **tn** represents the number of true negative predictions, **fp** represents the number of false positive predictions, and **fn** represents the number of false negative predictions.

Per Class Prediction’s Performance Based on Refernce Training Data.													
Dataset	Class Name	tp	tn	fp	fn	Accuracy	Specificity	Recall	Precision	Misclassification Rate	f1_Score (Dice)	Avg_iou	Std_ dev
Carotid Disease Patients	Artery Wall	637	73	0	43	0.94290	1.00000	0.93676	1.00000	0.05710	0.96735	0.76728	0.16596
Carotid Disease Patients	Hard Plaque	290	439	6	18	0.96813	0.98652	0.94156	0.97973	0.03187	0.96026	0.64748	0.19819
Carotid Disease Patients	Interior	622	105	2	24	0.96547	0.98131	0.96285	0.99679	0.03453	0.97953	0.93094	0.05395
Carotid Disease Patients	Soft plaque	444	280	15	14	0.96149	0.94915	0.96943	0.96732	0.03851	0.96838	0.75507	0.17420
Carotid Disease Patients	Thyroid	187	494	11	61	0.90438	0.97822	0.75403	0.94444	0.09562	0.83857	0.85365	0.18058
Carotid Disease Patients	All Classes											0.80296	0.14231

**Table 4 sensors-23-02806-t004:** Mean distances and stds for the aligned 3D US reconstructions.

Aligned 3D US Reconstructions	Mean Distance	Std Deviation	Scale	Theoretical Overlap	RMS
Patient 1 Scan1 + Scan2, CCOSS-Aligned	10.9656	25.5394	0.96326	100%	25.0231
Patient 1 Scan1 + Scan3, CCOSS-Aligned	24.5671	34.3423	0.943269	100%	16.064
Patient 1 Scan1 + Scan4, CCOSS-Aligned	5.03695	8.62778	1.01003	100%	25.2053
Patient 2 Scan1 + Scan2, CCOSS-Aligned	Corrupt data	Corrupt data	Corrupt data	Corrupt data	Corrupt data
Patient 2 Scan1 + Scan3, CCOSS-Aligned	7.62545	15.7469	0.628546	100%	50.2331
Patient 2 Scan1 + Scan4, CCOSS-Aligned	6.07168	16.0121	0.638966	100%	18.9179

## Data Availability

The datasets from the current study are available from the authors on reasonable request.

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
