# Peer review of "3D Ultrasound Reconstructions of the Carotid Artery and Thyroid Gland Using Artificial-Intelligence-Based Automatic Segmentation—Qualitative and Quantitative Evaluation of the Segmentation Results via Comparison with CT Angiography"

_sensors, 2023, doi:10.3390/s23052806_

Round 1

Reviewer 1 Report

Overall, the manuscript is well organized. I have the following comments to help the authors improve the manuscript:

1. Figure 3 needs to be improved. 

2. The format of table 2 needs to be revised. 

3. What do the symbols mean in table 3, such as tp, tn and fp? The authors need to explain them. 

4. How does the segmentation result compared with the reported models? The authors suggest to comparing with the previous reported models. 

Reviewer 2 Report

Authors developed AI-based automatic segmentation with ultrasound 3D imaging of the aortic nerve. The work is of interest and well-designed. Methods and results are scientifcally solid. However, the manuscript has to be modified before publication, especially the English enhancement. To be specific:

1. English writing needs to be enhanced. Typos and miss of words happen.

No punctuation in a super-long sentence “Because working in the 3D space and automatic segmentation will lower the operator dependency and using ultrasound imaging will be noninvasive a previously developed prototype for 3D ultrasound scans based on a standard ultrasound machine and a pose reading sensor was used for this study.”

What’s worse is this sentence is in the abstract!

“XXX Is a progressive disease”

2. The introduction should also provide information on other published ultrasound AI segmentation techniques.

3. A schematic illustration figure to demonstrate the ultrasound scanning method would be helpful.

4. Ultrasound 3D images are constructed from stacks of 2D images. How many frames per stack? What’s the step size between each 2D image? 

Reviewer 3 Report

This is a well-written description of a pilot study assessing the accuracy of several machine learning models to automatically segment ultrasound images of the carotid artery to quantify the presence of carotid plaques.  I have only a superficial understanding of machine learning, so I can only comment on the clinical aspects of this manuscript.  Overall, the clinical elements are well presented and there is clearly value in the work that the authors are pursuing to help increase the throughput and accuracy of carotid disease detection.  Notably, the study appeared to involve only data from 3 patients (1 normal and 2 abnormal) so the findings are, at best, hypothesis-generating preliminary data. 

My only constructive feedback is on the sentence below.

Line 61-62: “…while calcified plaque is 61 negatively associated with downstream ischemic events.”

This wording comes off as odd because “negatively associated” could mean an inverse relationship.  I think you mean “positively associated” or “correlated.”

Reviewer 4 Report

I think it is fine as it is.

However, we recommend modifying the layout before publishing. In particular, italicize subsections, and make sure indentation is consistent.

On line 235, "acquired" is surrounded by two "OBJ"s.

Round 2

Reviewer 1 Report

I do not have further technical comments for the authors. However, English problems and format problems can still be found in the manuscript. They should be revised before publishing. 

Author Response

Response to Reviewer 1 Comments

English language and style are fine/minor spell check required.

Thank you for your time and attention in completing this review. The authors carefully rechecked and made the necessary corrections. We hope that at this point the manuscript meets the standards of the reviewers and the editor.

Line 22 – 28: rephrased for clarity.

Line 81 – 87: rephrased for clarity.

Lines 96-99: rephrased for clarity.

Lines 104-107: rephrased for clarity.

Line 27: corrected spelling.

Lines 150-151: rephrased for clarity.

Lines 201-202: rephrased for clarity.

Line 309: corrected punctuation.

Line 319: rephrased for clarity.

Line 322: corrected punctuation.

Lines 330-332: rephrased for clarity.

Line 343: rephrased for clarity.

Lines 369-371: rephrased for clarity.
